# Homology Modelling, Molecular Docking and Molecular Dynamics Simulation Studies of CALMH1 against Secondary Metabolites of *Bauhinia variegata* to Treat Alzheimer’s Disease

**DOI:** 10.3390/brainsci12060770

**Published:** 2022-06-12

**Authors:** Noopur Khare, Sanjiv Kumar Maheshwari, Syed Mohd Danish Rizvi, Hind Muteb Albadrani, Suliman A. Alsagaby, Wael Alturaiki, Danish Iqbal, Qamar Zia, Chiara Villa, Saurabh Kumar Jha, Niraj Kumar Jha, Abhimanyu Kumar Jha

**Affiliations:** 1Institute of Biosciences and Technology, Shri Ramswaroop Memorial University, Barabanki 225003, Uttar Pradesh, India; noopur.khare2009@gmail.com (N.K.); sanjiv08@gmail.com (S.K.M.); 2Department of Biotechnology, Dr. A.P.J. Abdul Kalam Technical University, Lucknow 226021, Uttar Pradesh, India; 3Department of Pharmaceutics, College of Pharmacy, University of Hail, Hail 2240, Saudi Arabia; sm.danish@uoh.edu.sa; 4Department of Medical Laboratory Sciences, College of Applied Medical Sciences, Majmaah University, Majmaah 11952, Saudi Arabia; h.albadrani@mu.edu.sa (H.M.A.); s.alsaqaby@mu.edu.sa (S.A.A.); w.alturaiki@mu.edu.sa (W.A.); qamarzia@mu.edu.sa (Q.Z.); 5Health and Basic Sciences Research Center, Majmaah University, Al Majmaah 15341, Saudi Arabia; 6School of Medicine and Surgery, University of Milano-Bicocca, 20900 Monza, Italy; chiara.villa@unimib.it; 7Department of Biotechnology, School of Engineering and Technology, Sharda University, Greater Noida 201310, Uttar Pradesh, India; Saurabh.jha@sharda.ac.in (S.K.J.); nirajkumarjha2011@gmail.com (N.K.J.); 8Department of Biotechnology, School of Applied & Life Sciences (SALS), Uttaranchal University, Dehradun 248007, Uttarakhand, India; 9Department of Biotechnology Engineering and Food Technology, Chandigarh University, Mohali 140413, Punjab, India

**Keywords:** homology modelling, LOMETS, MUSTER, iGEMDOCK, AutoDock vina

## Abstract

Calcium homeostasis modulator 1 (CALHM1) is a protein responsible for causing Alzheimer’s disease. In the absence of an experimentally designed protein molecule, homology modelling was performed. Through homology modelling, different CALHM1 models were generated and validated through Rampage. To carry out further in silico studies, through molecular docking and molecular dynamics simulation experiments, various flavonoids and alkaloids from *Bauhinia variegata* were utilised as inhibitors to target the protein (CALHM1). The sequence of CALHM1 was retrieved from UniProt and the secondary structure prediction of CALHM1 was done through CFSSP, GOR4, and SOPMA methods. The structure was identified through LOMETS, MUSTER, and MODELLER and finally, the structures were validated through Rampage. *Bauhinia variegata* plant was used to check the interaction of alkaloids and flavonoids against CALHM1. The protein and protein–ligand complex were also validated through molecular dynamics simulations studies. The model generated through MODELLER software with 6VAM A was used because this model predicted the best results in the Ramachandran plot. Further molecular docking was performed, quercetin was found to be the most appropriate candidate for the protein molecule with the minimum binding energy of −12.45 kcal/mol and their ADME properties were analysed through Molsoft and Molinspiration. Molecular dynamics simulations showed that CALHM1 and CALHM1–quercetin complex became stable at 2500 ps. It may be seen through the study that quercetin may act as a good inhibitor for treatment. With the help of an in silico study, it was easier to analyse the 3D structure of the protein, which may be scrutinized for the best-predicted model. Quercetin may work as a good inhibitor for treating Alzheimer’s disease, according to in silico research using molecular docking and molecular dynamics simulations, and future in vitro and in vivo analysis may confirm its effectiveness.

## 1. Introduction

Alzheimer’s disease is a long-term illness that causes brain cell loss and degeneration. The most common kind of dementia is Alzheimer’s disease, which is characterised as a gradual loss of mental, communicative, and social abilities that makes it difficult for a person to operate independently [1]. Current Alzheimer’s disease medicines may temporarily alleviate symptoms or delay the progression of the illness. Medications can often assist patients with Alzheimer’s disease enhance their neuron function. For those with Alzheimer’s disease, a variety of programs and services can be quite beneficial [2].

Intracellular calcium (Ca^2+^) dynamics govern key neuronal functions such as neurotransmission, synaptic plasticity, learning, and memory, and signalling cascades, cytoskeleton modifications, synaptic function, and neuronal survival are all affected by changes in Ca^2+^ dynamics [3]. Several investigations have indicated the essential role of Ca^2+^ dysregulation in central Alzheimer’s disease-related pathogenic processes since the first systematic hypothesis was proposed twenty years ago (AD). Disturbances in Ca^2+^ signals were discovered in the early stages of Alzheimer’s disease, even before the build-up of amyloid ß-peptide (Aß), a clinical marker of the disease [4]. A growing body of evidence shows that mutations in AD-related genes such as presenilins, amyloid precursor protein, or apolipoprotein-E affect Ca^2+^ signalling, leading to apoptosis, synaptic plasticity failure, and neurodegeneration [5].

Ca^2+^o (extracellular calcium) plays an important part in physiological processes. In a number of physiological and pathological circumstances, changes in Ca^2+^o concentration ([Ca^2+^o]) have been discovered to modify neuronal excitability, although the mechanisms by which neurons detect [Ca^2+^]o remain unknown [6]. Calcium homeostasis modulator 1 (CALHM1) expression has been shown to generate cation currents in cells and enhance the concentration of cytoplasmic Ca^2+^ ([Ca^2+^]) in response to Ca^2+^o removal and subsequent addition. It is unclear if CALHM1 is a pore-forming ion channel or a modulator of endogenous ion channels. CALHM1 is also expressed in mouse cortical neurons, which respond to reduced [Ca^2+^]o with increased conductivity and potential firing action, as well as a significantly higher [Ca^2+^i] when Ca^2+^o is withdrawn [7]. Those reactions, on the other hand, are significantly reduced in mouse neurons that have had CALHM1 genetically eliminated. These findings demonstrate that CALHM1 is an evolutionarily conserved family of ion channels that senses membrane voltage and external Ca^2+^ levels and plays a role in cortical neuronal excitability and Ca^2+^ homeostasis, notably in response to decreasing and restoring [Ca^2+^o] [8].

The absence of an experimentally characterized structure has hampered progress in determining the function of CALHM1 in Alzheimer’s disease. The two most common experimental approaches for determining the structure of proteins are X-ray crystallography and nuclear magnetic resonance (NMR) spectroscopy. These techniques, however, have requirements such as a high time and personnel needs [9]. Obtaining protein sequences, on the other hand, is much easier than obtaining protein structure, thanks to current sequencing methods. As a result, databases such as UniProt (https://www.uniprot.org/ (accessed on 7 January 2021)) and TrEMBL (Translated EMBL) (https://www.uniprot.org/statistics/TrEMBL (accessed on 8 January 2021)) include many protein sequences. In the late twentieth century, computational approaches for predicting the structure of proteins gave a sequence of amino acids. The information essential for a protein’s correct folding is encoded in its amino acid sequence, according to research (Anfinsen’s dogma). Homology modelling (based on sequence comparison) and threading are presently the most used computational approaches for predicting protein structure (based on sequence comparison) [10].

The goal of this research was to create useful models of the CALHM1 computational protein structure. As a result, additional research and analysis of CALHM1 function in Alzheimer’s disease will be aided [11]. Comparative modelling was carried out in the absence of its experimentally deduced structure using the software programs MODELLER (https:/salilab.org/modeller/ (accessed on 15 January 2021)), LOMETS (Local MetaThreading Server) (https:/zhanglab.ccmb.med.umich.edu/LOMETS/ (accessed on 19 January 2021)) and MUSTER (MUlti-Sources ThreadER) (https:/zhanglab.cccmb.med.umich.edu/MUSTER/ (accessed on 30 January 2021)) [12]. RAMPAGE (Ramachandran Plot Assessment) (http:/mordred.bioc.cam.ac.uk/~rapper (accessed on 3 February 2021)) was then used to test the model structure. Using SPDBV (Swiss PDB Viewer) (https:/spdbv.vital-it.ch/ (accessed on 5 February 2021)) software, the energy minimisation of the four modelled structures was carried out. Using GOR4 (Garnier–Osguthorpe–Robson) (https:/npsa-prabi.ibcp.fr/NPSA/npsa gor4.html (accessed on 5 February 2021)), CFSSP (Chou and Fasman Secondary Structure Prediction Server) (http:/www.biogem.org/tool/chou-fasman/ (accessed on 5 February 2021)), and SOPMA (Self-Optimized Prediction System with Alignment) algorithms (https:/npsa-prabi.ibcp.fr/cgi-bin/npsa automat.pl?page=/NPSA/npsa sopma.html (accessed on 5 February 2021)) also generated secondary protein structure [13].

## 2. Material and Methods

The protein structure prediction modelling for comparative modelling consisted of the following steps. Target identification came first, followed by alignments of the target and prototype sequences. The model was built when the alignment template procedure was completed. Finally, the model’s strength, stearic collisions, and stability were evaluated.

### 2.1. Protein Sequence Retrieval

The CALHM1 protein sequence (accession number: Q8IU99 (CAHM1_HUMAN) was saved from the UniProt database (https://www.uniprot.org/ (accessed on 7 January 2021)) [14].

### 2.2. Protein Secondary Structure Prediction

The CALHM1 protein sequence (accession number: O43315 (AQP9 HUMAN)) was further subjected to secondary structure prediction on the Expasy server using GOR4, SOPMA and CFSSP [15].

### 2.3. Protein Tertiary Structure Prediction through Template Identification

A thorough search of the PDB (Protein Data Bank) (http://www.rcsb.org/ (accessed on 16 February 2021)) was conducted to search the most similar sequences already known for experimentally designed structures. The template protein structures (6VAM A and 6LMT A) were analysed as the most accurate template for identifying the three-dimensional protein structure based on various factors such as E-value, percentage identity, alignment score, and query coverage [16].

### 2.4. Modelling

The protein CALHM1 three-dimensional structure was calculated using MODELLER version 9.15, LOMETS, and the MUSTER server. MODELLER carries out a comparative modelling of the proteins according to the identified template. LOMETS is based on a metathreading technique for identifying the protein structure based on a template. MUSTER is based on a protein threading algorithm, which identifies PDB library template structures [17]. It generates sequence–template alignments with multiple structural data by combining different sequences. Several models were created through two templates, and a comparison of their DOPE score was selected for the best model [18].

### 2.5. Validation of the Structure

The RAMPAGE server was used to create Ramachandran plots in order to validate the predicted protein structures by looking at criteria such as preferred, allowed, and outside amino acid residue areas. The pdb files of the best target gene models predicted by MODELLER, LOMETS, and MUSTER were sent to the RAMPAGE service to create Ramachandran plots. Plots were identified for the anticipated structures, and the plots were matched to determine the best structure among the projected structures and further studies. Validation of the protein structure’s quality was also carried out using the ProSA server [12].

### 2.6. Energy Minimisation of the Predicted Molecule

For SPDBV to achieve the lowest energy conformation, the identified and analysed models were subjected to energy minimisation.

### 2.7. Preparation of Ligand Molecule

The flavonoids and alkaloids structure of the plant *Bauhinia variegata* were retrieved in sdf format from the PubChem online database. The stem bark of the plant contains beta-carotene, quercetin, stigmasterol, hentriacontane, flavanone, isoquericetroside, kaempferol-3-glucoside, lupeol, myricetol, phenanthriquinone, quercitroside, rutoside, xanthophyll, dihydroquercetin, octacosanol, and beta-sitosterol. All the structures were retrieved in 3D structure in SDF format and were further converted into pdb format through online converting tool [19].

### 2.8. Initial Docking through iGEMDOCK Software

Initial docking was performed to screen the ligands on the basis of the binding energy. The docking process was carried out through iGEMDOCK version 2.1. The result was in the form of an electrostatic force, hydrogen bonds, and Van Der Waals forces [20]. The docking was performed between protein and ligand with a population size of 200 and the number of generations was 70 with 2 solutions [21].

### 2.9. Final Molecular Docking through AutoDock Vina and Drug Likeliness Property Analysis

The ligands were screened through iGEMDOCK and these screened ligands were tested against CALHM1 protein through AutoDock vina software. This software tool is freely available online. The protein CALHM1 was assigned with Kollman charges and polar hydrogens. The screened ligands were added with nonpolar hydrogen atoms and with Gasteiger partial charges. The torsion angles were allowed to rotate freely. A grid box of 80 × 80 × 80 Å was adjusted in such a manner that it was covering the target molecule to give the best docked result. The docking algorithm was adjusted to 100 runs. The default parameters were the Lamarckian genetic algorithm (LGA) and the empirical free energy function. The best-targeted molecule was screened further based on its minimal binding energy (Kcal/mol) [22].

Drug likeliness analysis was done through Molsoft (http://www.molsoft.com/ (accessed on 2 March 2021)) and Molinspiration (http://www.molinspiration.com/ (accessed on 2 March 2021)). Different properties of the screened ligand were analysed through pkCSM [23].

### 2.10. Molecular Dynamics Simulations

According to the molecular docking results, a molecular dynamics simulation was performed. The molecule which showed the minimum binding energy with the protein molecule was compared with the protein molecule for the dynamics study. The Groningen Machine for Chemical Simulations (GROMACS) 4.5.6 package was used to run molecular dynamics simulations. For the simulations, Gromacs was utilised to build the protein target and ligand file. With the help of an online server PRODRG2.5, the topology parameters of the ligand were generated [24]. The protein and ligand complex was placed inside the shell. The volume of the box was 284.14 nm^3^ and the distance between the protein molecule and the box was kept at 1.0 nm. After adding 8 sodium ions to the shell, simple point charges and water molecules were neutralized. Energy minimisation was achieved using the steepest approach of 8 ps. The machine was balanced at 40 ps when the temperature was raised to 300 K. The simulations at 10 ns were performed at 1 bar and at the temperature of 300 K. Finally, an all-bond restriction was employed to keep the ligand from migrating during molecular dynamics [25].

## 3. Results and Discussion

### 3.1. Protein Sequence 

The protein sequence of CALHM1 was retrieved in FASTA format from the UniProt database as shown in Figure 1.

### 3.2. Protein Secondary Structure Prediction

The secondary structure prediction of CALHM1 was carried out with the aid of methods such as Chou and Fasman Secondary Structure Prediction Server (CFSSP), Garnier–Osguthorpe–Robson (GOR4), and the Self-Optimised Prediction Method with Alignment (SOPMA). Information from GOR4, CFSSP, and SOPMA Expasy tools were obtained and the secondary structures such as alpha helix, beta strand, and random coil for the target CALHM1 were extracted.

Chou and Fasman Secondary Structure Prediction Server (CFSSP) is an empirical predictive tool of secondary protein structures. The method depends on analyses of the relative frequencies in alpha helices, beta sheets, and turns of each amino acid based on known protein structures solved with X-ray crystallography. The analysis of the CFSSP showed that CALHM1 consisted of 271 alpha helix, 248 extended strands, and 34 turns, as shown in Figure 2.

The Garnier–Osguthorpe–Robson (GOR4) method is a technique based on information theory to predict the secondary structures in proteins. It uses probability factors derived from empirical research of known tertiary protein structures solved using X-ray crystallography. The analysis of GOR4 showed that CALHM1 consisted of 133 alpha helix, 45 extended strands, and 168 random coils, as shown in Figure 3.

The Self-Optimised Prediction Method with Alignment (SOPMA) is an Expasy server protein-aided secondary structure prediction tool. Using consensus estimation from several alignments, the algorithm contributes to significant advances in protein secondary structure. Analysing SOPMA, CALHM1 consisted of 182 alpha helix, 34 extended strands, and 119 random coils as shown in Figure 4.

### 3.3. Template Identification

PDB Blast was performed to define CALHM1 modelling prototype structures for comparative homology modelling. We compared the templates and selected two of them (6VAM A and 6LMT A) based on their query cover, E-value and identity as shown in Table 1. Using MODELLER software, the two structures were downloaded from PDB for modelling the protein.

### 3.4. Modelling through MODELLER

Through 6VAM A and 6LMT A prototype files, structures were modelled using MODELLER version 9.15 software for the protein CALHM1. Fifty models were produced using 6VAM A and 6LMT A modellers. With the help of the DOPE score as a criterion, we selected one best model for 6VAM A (Model 1) as shown in Figure 5 and Figure 6LMT A (Model 2) as shown in Figure 6.

### 3.5. Structure Prediction through LOMETS Server

LOMETS server was used as a meta threading approach to identify the 3D structure of the given sequence. Ten protein structures were generated, and the best structures were further evaluated. Comparing the Z-score and maximum coverage, the best model among them was selected as shown in Figure 7.

### 3.6. Protein Structure Prediction Using MUSTER Server

In addition, the MUSTER online server was used for protein treading. This server created ten different protein sequence models, among which the structure with the lowest Z-score and the maximum coverage was chosen as the fittest structure as shown in Figure 8.

### 3.7. Structure Validation Using Ramachandran Plot

The selected four protein structures were uploaded to RAMPAGE to analyse the predicted structures, which produced the Ramachandran plots for the predicted protein structures as shown in Figure 9. In this plot, the amino acids (residues) have been identified in three distinct regions. The three distinct regions were favoured region, allowed region, and outlier region as shown in Table 2. Further structure validation was also performed through the ProSA server (https://prosa.services.came.sbg.ac.at/prosa.php (accessed on 3 February 2021)) to analyse the protein structure as shown in Figure 10.

### 3.8. Energy Minimisation

The energy minimisation was carried through SPDBV (Swiss PDB Viewer) (https://spdbv.vital-it.ch/ (accessed on 5 February 2021)) software for the predicted models. The values retrieved from the energy minimisation were analysed to identify the best protein structure which was predicted as shown in Table 3. MODELLER 6VAM A had the least energy content (E = 2468.876 KJ/mol).

### 3.9. Ligands from Bauhinia variegata

The three-dimensional structure in sdf format were downloaded from PubChem. The secondary metabolites were xanthophyll, beta-carotene, beta-sitosterol, dihydroquercetin, quercetin, stigma sterol, hentriacontane, octacosanol, flavanone, isoquericetroside, kaempeferol-3-glucoside, lupeol, myricetol, phenanthriquinone, quercitroside, and rutoside, as shown in Table 4.

### 3.10. Screening of Ligands through iGEMDOCK 

All the downloaded ligands were interacted with the protein molecule (CALHM1). Table 5 shows how the Van der Waal forces, binding energy, and hydrogen bond were used to filter the best docked molecules. The structure of the screened docked protein– ligands are shown in Figure 11.

### 3.11. Molecular Docking Analysis through AutoDock Vina

Using the AutoDock vina program, the screened ligands were docked with CALHM1. The ligands were then sorted according to their minimal binding affinity. The optimum posture was determined based on the lowest binding affinity, as illustrated in Figure 12. Quercetin (CID: 5280343) exhibited the lowest binding affinity with CALHM1 according to molecular docking findings. The optimum energy value after comparing the poses of quercetin with CALHM1 was −12.45 Kcal/mol, as indicated in Table 6. The AutoDock vina energies were evaluated.

### 3.12. Cheminformatics Properties and Lipinski’s Rule of Five Validation of Quercetin

The cheminformatics properties were studied for quercetin. The properties were molecular formula, molecular weight (g/mol), molar refractivity (cm^3^), density (cm^3^), drug likeness, etc., as shown in Table 7. The molecular weight of quercetin was 302.24 g/mol, the value of log P was 0.56, molecular refractivity was 122.60 cm^3^. The comparative result of quercetin predicts it to be a good candidate. Quercetin was also validated for its Lipinski’s rule of five properties, the value of quercetin predicted <10 hydrogen bond acceptors, <5 hydrogen bond donors, <500 g/mol molecular weight, <5 log P value [26].

### 3.13. Quercetin’s Pharmacokinetic Properties

Properties such as absorption, distribution, metabolism, excretion and toxicity properties (ADMET) were analysed for quercetin under pharmacokinetics. 

The absorption property was analysed by intestinal water solubility at −2.925 log mol/L, and the intestinal solubility was found to be 96.902 percent, and the skin permeability value was −2.735 log Kp, which showed a strong quercetin structure that validated its good behaviour in terms of drug likeness. The blood brain barrier (BBB) and central nervous system (CNS) permeability values of quercetin were analysed for distribution properties with a weak BBB value of −1.098 log BB. However, the permeability value of the central nervous system (CNS) was −3.065 log PS. The CYP3A4 substrate, which is the isoform of cytochrome P450, confirmed the metabolism property. The property of excretion showed that the total clearance value was 0.407, which showed that quercetin had nontoxic actions, and nontoxicity was inferred [27]. Table 8 shows all of the results.

### 3.14. Molecular Dynamic Simulations Analysis

The protein target (CALHM1) along with quercetin and CALHM1 were selected for the molecular dynamics simulations to check the conformations. Using the GROMACS, the trajectories were analysed in terms of RMSD (root mean square deviation), Rg (radius of gyration), SASA (solvent accessible surface area), and RMSF (root mean square fluctuation).

### 3.15. Root Mean Square Deviation (RMSD)

The RMSD was used to identify the stability of unliganded CALHM1 and CALHM1 with quercetin. The system was in balance, with RMSD fluctuating about 2500 ps. As seen in Figure 13, the backbone atoms grew up to around 0.23 nm before stabilising until the end of the simulation, showing that the molecular system was then properly set.

### 3.16. Radius of Gyration 

After around 2500 ps, the systems stabilised, indicating that the molecular dynamics simulation equilibrated. The CALHM1–quercetin binding was anticipated to be excellent based on the radius of gyration. As illustrated in Figure 14, the environment does not alter during contact.

### 3.17. Solvent Accessible Surface Area (SASA)

The CALHM1 total solvent exposed surface area was displayed at 10,000 ps. The differences seen in the SASA graph were a bit similar to those of the radius of gyration. As demonstrated in Figure 15, it can be seen that there is a similarity between the SASA and the radius of gyration, which shows the accuracy in the simulation results. 

### 3.18. Root Mean Square Fluctuation (RMSF)

The RMSF was used to investigate the mobility of CALHM1 residues in the presence and absence of ligands. The findings show that variations greater than 0.25 nm indicate residues located far from each ligand’s binding sites. Furthermore, as seen in Figure 16, the residues in contact with the quercetin were the most stable and had the lowest RMSF values.

## 4. Conclusions

In recent years, there has been a huge growth in the use of various software and algorithms to predict protein structure using in silico methods. However, the precision of structure prediction, the severity of fold assignment errors, and the modelling of side chains and loops all need a significant body shift. CALHM1 (calcium homeostasis modulator 1) is involved in the pathogenesis of Alzheimer’s disease. CALHM1’s various models were developed and verified using homology modelling and Rampage. The interaction of alkaloids and flavonoids with CALHM1 was tested using the *Bauhinia variegata* plant. The model created with MODELLER software and 6VAM A was chosen because it performed best on the Ramachandran plot. Quercetin was shown to be the best choice for the protein molecule, with a minimum binding energy of −12.45 kcal/mol, and its ADME qualities were assessed using Molsoft and Molinspiration. At 2500 ps, CALHM1 and the CALHM1–quercetin combination became stable, according to molecular dynamics simulations. Finally, the in silico analysis suggested that quercetin might be a suitable therapeutic inhibitor. Quercetin may operate as a good inhibitor for treating Alzheimer’s disease, according to in silico research using molecular docking and molecular dynamics simulations, and future in vitro and in vivo investigations may establish its therapeutic potential.

## Figures and Tables

**Figure 1 brainsci-12-00770-f001:**
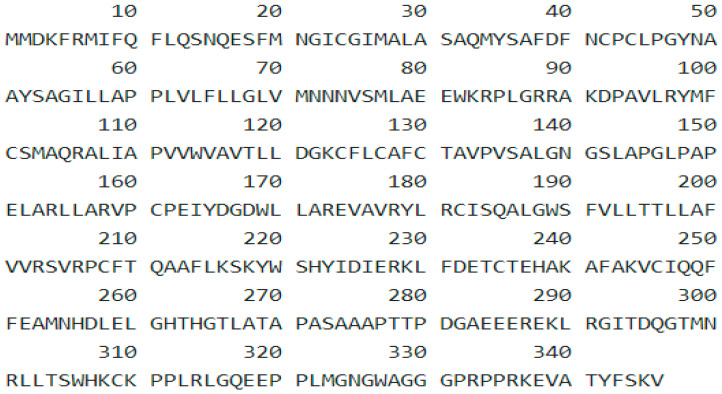
Protein sequence of CALHM1.

**Figure 2 brainsci-12-00770-f002:**
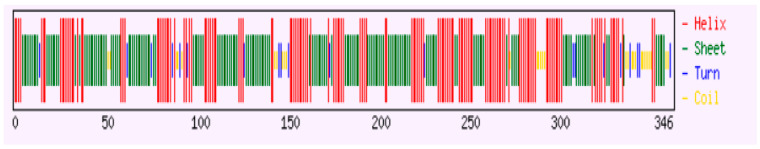
CFSSP result.

**Figure 3 brainsci-12-00770-f003:**
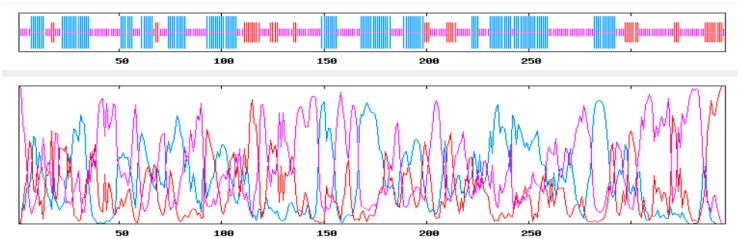
GOR4 result (dark blue is denoting alpha helix, light blue is denoting pi helix, dark pink is denoting beta bridge, red is denoting extended strands).

**Figure 4 brainsci-12-00770-f004:**
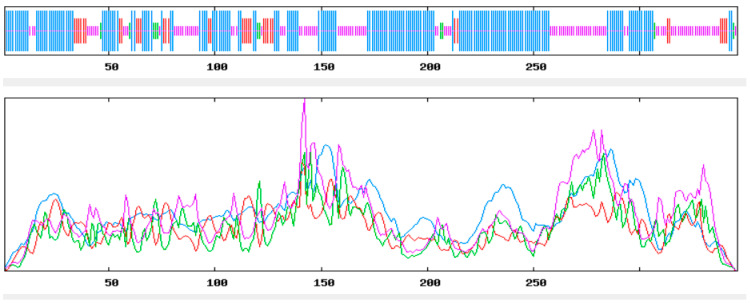
SOPMA result (dark blue is denoting alpha helix, green is denoting pi helix, dark pink is denoting beta bridge, red is denoting extended strands).

**Figure 5 brainsci-12-00770-f005:**
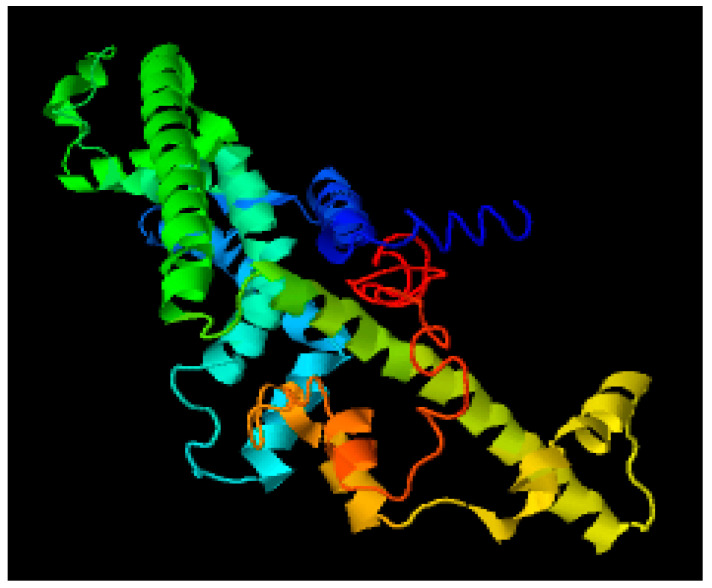
Best predicted model by MODELLER (6VAM A).

**Figure 6 brainsci-12-00770-f006:**
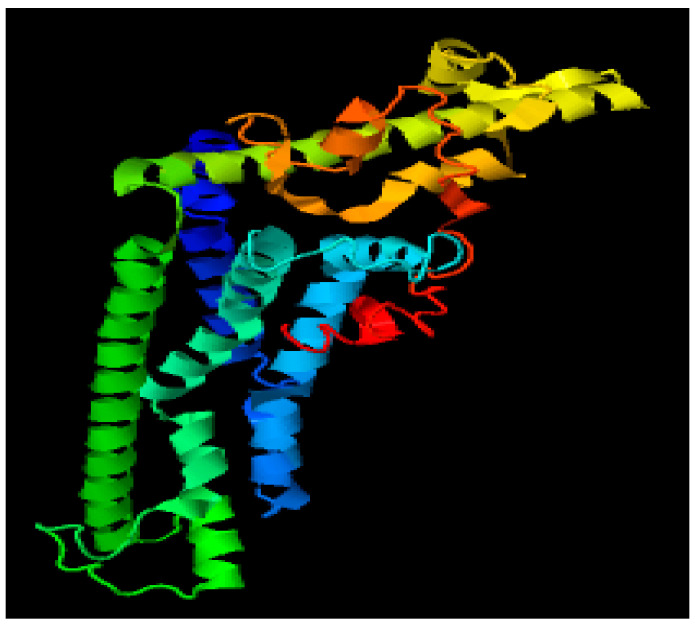
Best predicted model by MODELLER (6LMT A).

**Figure 7 brainsci-12-00770-f007:**
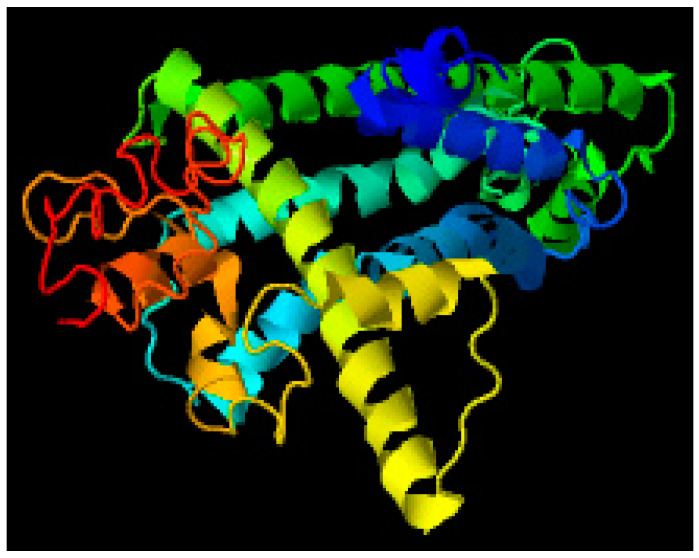
Best predicted model by LOMETS.

**Figure 8 brainsci-12-00770-f008:**
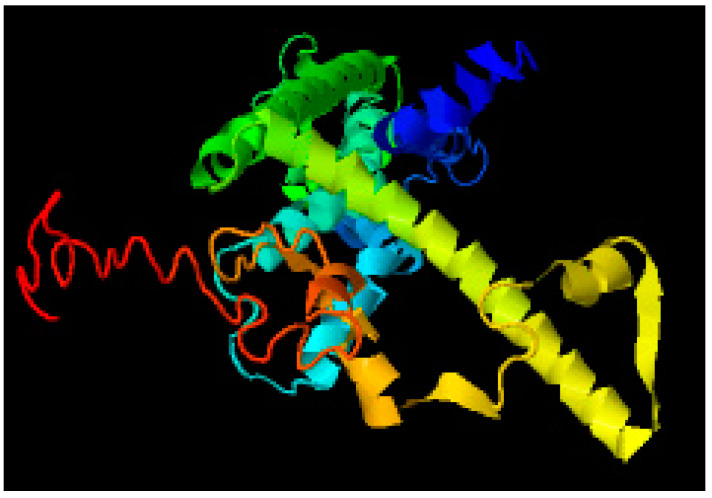
Best predicted model by MUSTER.

**Figure 9 brainsci-12-00770-f009:**
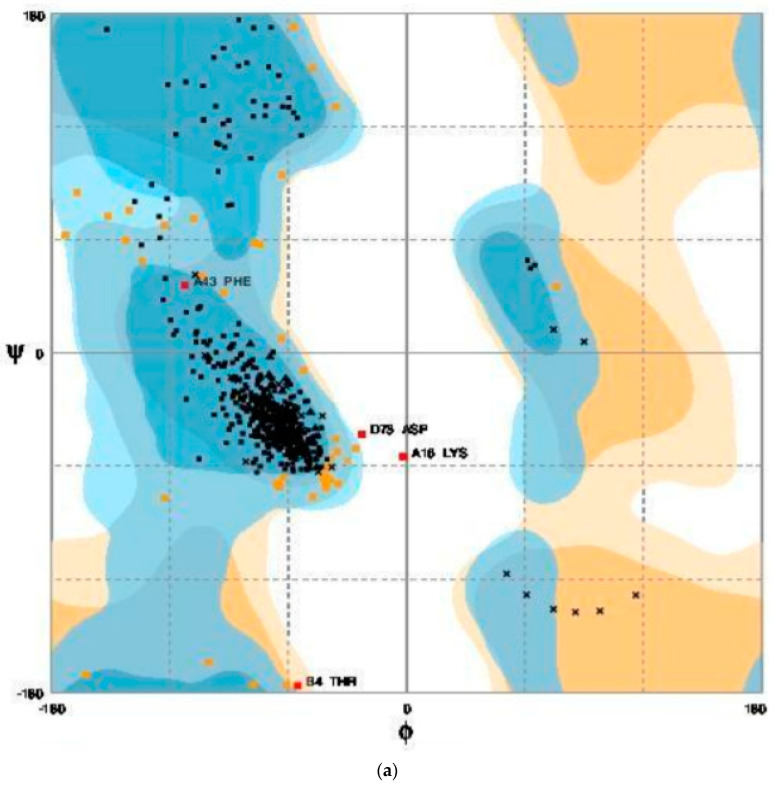
Ramachandran plots for (**a**) MODELLER (6VAM A); (**b**) MODELLER (6LMT A); (**c**) LO-ETS server; (**d**) MUSTER server.

**Figure 10 brainsci-12-00770-f010:**
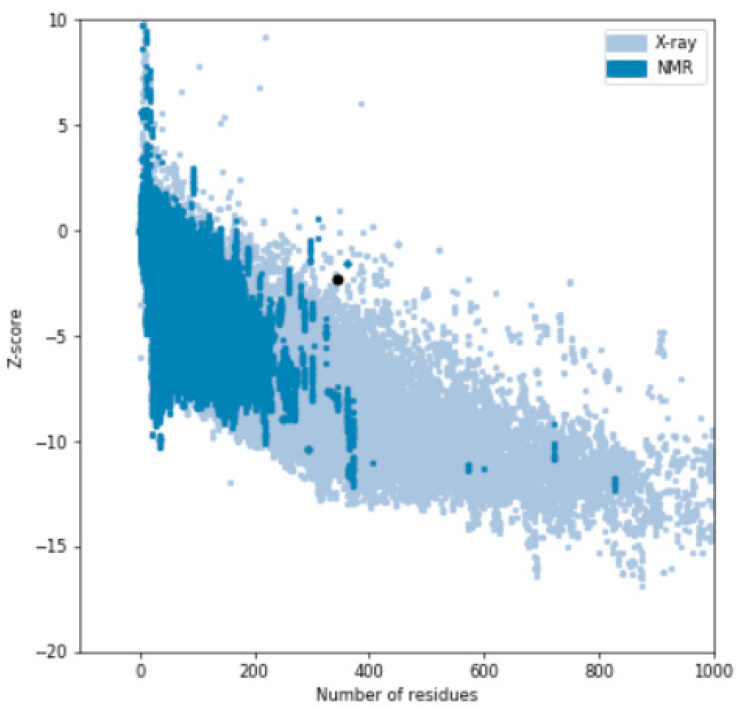
Quality of the protein structure (ProSA Server).

**Figure 11 brainsci-12-00770-f011:**
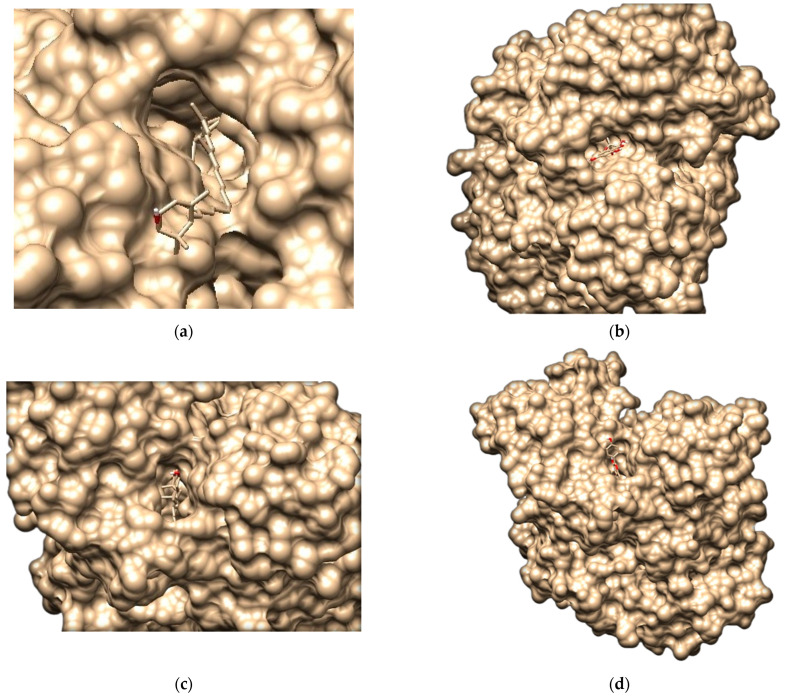
Molecular docking analysis: (**a**) pose view of CALHM1 with betacarotene; (**b**) pose view of CALHM1 with betasitosterol; (**c**) pose view of CALHM1 with quercetin; (**d**) pose view of CALHM1 with stigmasterol; (**e**) pose view of CALHM1 with xanthophyll; (**f**) pose view of CALHM1 with dihydroquercetin.

**Figure 12 brainsci-12-00770-f012:**
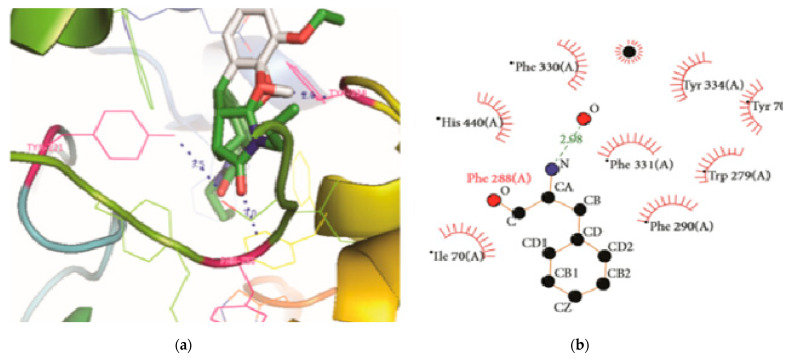
CALHM1 docking and Ligplot interaction with quercetin. (**a**) The hydrogen bond distance between the docked ligand and the active site is shown; (**b**) a two-dimensional depiction of a ligand and a protein residue.

**Figure 13 brainsci-12-00770-f013:**
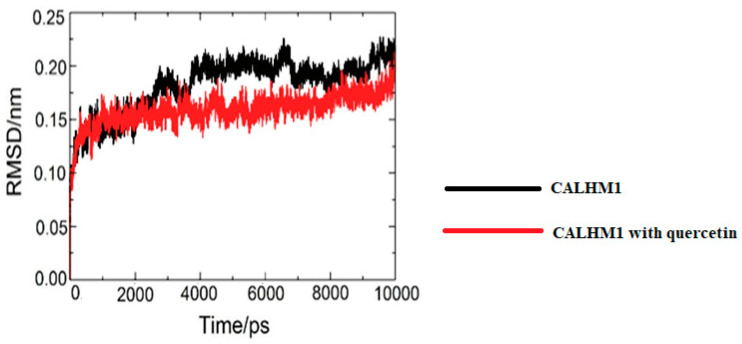
Time dependence of root mean square deviation. RMSD values for unliganded CALHM1 and CALHM1–quercetin complex.

**Figure 14 brainsci-12-00770-f014:**
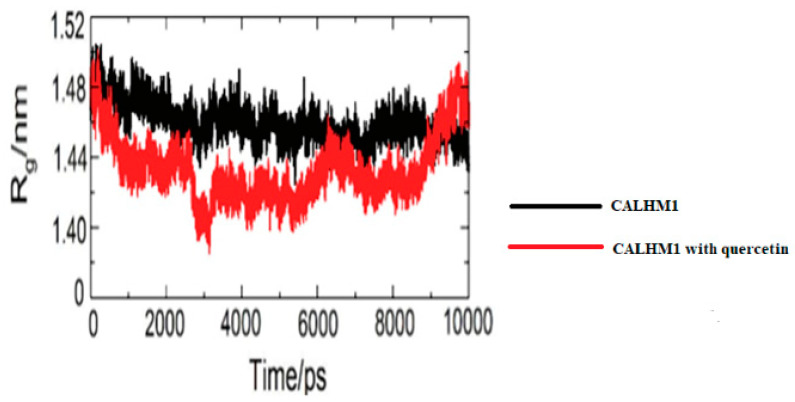
Radius of gyration (Rg) during 10,000 ps of MD simulation of unliganded CALHM1 and CALHM1–quercetin complex.

**Figure 15 brainsci-12-00770-f015:**
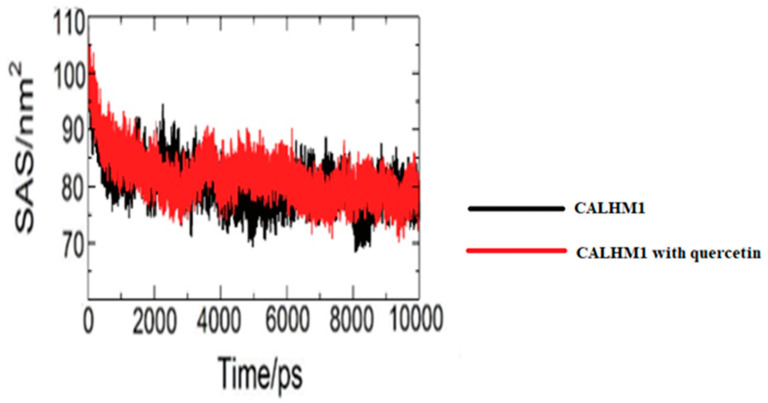
Solvent accessible surface area (SASA) during 10,000 ps of MD simulation of unliganded CALHM1 and CALHM1–quercetin complex.

**Figure 16 brainsci-12-00770-f016:**
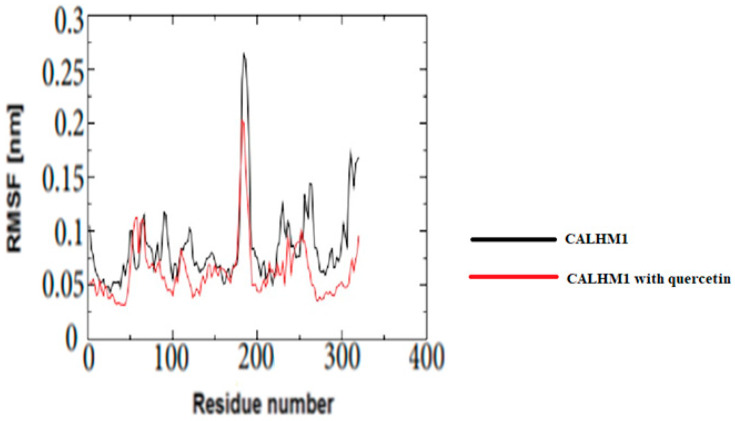
The RMSF values of unliganded CALHM1 and CALHM1–quercetin complex.

**Table 1 brainsci-12-00770-t001:** CALHM1 BLAST parameters.

Query Cover	E-Value	Identity	Accession
99%	6 × 10^−169^	68.36%	6VAM A
88%	4 × 10^−139^	58.82%	6LMT A

**Table 2 brainsci-12-00770-t002:** Ramachandran plot for each model.

	MODELLER		LOMETS	MUSTER
	6VAM A	6LMT A		
Favoured region	187	172	177	196
Allowed region	5	6	14	10
Outlier region	2	3	15	6

**Table 3 brainsci-12-00770-t003:** Energy minimisation values through SPDBV.

	MODELLER		LOMETS	MUSTER
	6VAM A	6LMT A		
Energy (KJ/mol)	2468.876	5688.255	10,265.889	8714.236

**Table 4 brainsci-12-00770-t004:** Compounds and their respective PubChem IDs.

Compound	PubChem ID
Hentriacontane	CID: 12410
Octacosanol	CID:68406
Stigmasterol	CID:5280794
Betasitosterol	CID:222284
Flavanone	CID:10251
Isoquericetroside	CID:5484006
Kaempeferol-3-glucoside	CID:6325460
Lupeol	CID:259846
Myricetol	CID:5281672
Phenanthriquinone	CID:6763
Quercitroside	CID:5280459
Rutoside	CID:5280805
Xanthophyll	CID:5281243
Beta- carotene	CID:5280489
Dihydroquercetin	CID:439533
Quercetin	CID:5280343

**Table 5 brainsci-12-00770-t005:** Initial docking by iGEMDOCK.

Ligands	Binding Energy	VDW	HBond
Quercetin (CID: 5280343)	−12.66	−22.13	−2.34
Dihydroquercetin (CID: 439533)	−10.30	−21.11	−2.18
Beta-carotene (CID: 5280489)	−10.26	−20.11	−3.42
Xanthophylls (CID: 5281243)	−8.20	−11.33	−4.57
Stigma sterol (CID: 5280794)	−7.80	−29.20	−7.6
Beta-sitosterol (CID: 222284)	−6.70	−30.29	−3.41

**Table 6 brainsci-12-00770-t006:** Docking result of quercetin against CALHM1.

S.No	Mode	Affinity (kcal/mol)	Distance From Best Mode RMSD l.b	Distance From Best Mode RMSD u.b
1.	1	−12.45	0.000	0.000
2.	2	−12.34	21.115	20.357
3.	3	−12.12	12.235	12.514
4.	4	−11.65	9.656	6.524
5.	5	−11.25	8.459	2.722
6.	6	−10.81	8.287	10.650
7.	7	−9.45	7.775	1.089
8.	8	−9.32	6.002	11.924
9.	9	−8.23	6.028	14.615

**Table 7 brainsci-12-00770-t007:** Cheminformatics properties of quercetin.

Molecular Formula	C_15_H_10_O_7_
Molecular weight (g/mol)	302.24
Hydrogen bond acceptor	7
Hydrogen bond donor	5
Rotatable bonds	1
Log *p*	0.56
No of atoms	22
Polar surface area (A^2^)	103.49 A^2^
Molar refractivity (cm^3^)	122.60
Density (cm^3^)	1.23
Molar volume (cm^3^)	268.73 cm^3^
Drug likeness	1
Lipinski validation	yes
GPCR ligand	−0.06
Ion channel modulator	−0.19
Kinase inhibitor	0.28
Nuclear receptor ligand	0.36
Protease inhibitor	−0.25
Enzyme inhibitor	0.28

**Table 8 brainsci-12-00770-t008:** Pharmacokinetic properties of quercetin.

S.No.	Property	Model Name	Predicted Value	Unit
1.	Absorption	Water solubility	−2.925	Numeric (log mol/L)
2.	Absorption	Caco_2_ permeability	−0.229	Numeric (log Papp in 10–6 cm/s)
3.	Absorption	Intestinal absorption (human)	96.902	Numeric (% absorbed)
4.	Absorption	Skin permeability	−2.735	Numeric (log Kp)
5.	Absorption	P-glycoprotein substrate	Yes	Categorical (Yes/No)
6.	Absorption	P-glycoprotein I inhibitor	No	Categorical (Yes/No)
7.	Absorption	P-glycoprotein II inhibitor	No	Categorical (Yes/No)
8.	Distribution	VDss (human)	1.559	Numeric (log L/kg)
9.	Distribution	Fraction unbound (human)	0.206	Numeric (Fu)
10.	Distribution	BBB permeability	−1.098	Numeric (log BB)
11.	Distribution	CNS permeability	−3.065	Numeric (log PS)
12.	Metabolism	CYP2D6 substrate	No	Categorical (yes/no)
13.	Metabolism	CYP3A4 substrate	No	Categorical (yes/no)
14.	Metabolism	CYP1A2 inhibitor	Yes	Categorical (yes/no)
15.	Metabolism	CYP2C19 inhibitor	No	Categorical (yes/no)
16.	Metabolism	CYP2C9 inhibitor	No	Categorical (yes/no)
17.	Metabolism	CYP2D6 inhibitor	No	Categorical (yes/no)
18.	Metabolism	CYP3A4 inhibitor	No	Categorical (yes/no)
19.	Excretion	Total clearance	0.407	Numeric (log ml/min/kg)
20.	Excretion	Renal OCT2 substrate	No	Categorical (yes/no)
21.	Toxicity	AMES toxicity	No	Categorical (yes/no)
22.	Toxicity	Max. tolerated dose (human)	0.499	Numeric (log mg/kg/day)
23.	Toxicity	hERG I inhibitor	No	Categorical (yes/no)
24.	Toxicity	hERG II inhibitor	No	Categorical (yes/no)
25.	Toxicity	Oral rat acute toxicity (LD50)	2.471	Numeric (mol/kg)
26.	Toxicity	Oral rat chronic toxicity (LOAEL)	2.612	Numeric (log mg/kg_bw/day)
27.	Toxicity	Hepatotoxicity	No	Categorical (yes/no)
28.	Toxicity	Skin sensitisation	No	Categorical (yes/no)
29.	Toxicity	*T. pyriformis* toxicity	0.288	Numeric (log μg/L)
30.	Toxicity	Minnow toxicity	3.721	Numeric (log mM)

## Data Availability

Not applicable.

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
