# Peer review of "Homology Modelling, Molecular Docking and Molecular Dynamics Simulation Studies of CALMH1 against Secondary Metabolites of Bauhinia variegata to Treat Alzheimer’s Disease"

_brainsci, 2022, doi:10.3390/brainsci12060770_

Round 1

Reviewer 1 Report

Khare et al. written interesting and important article entitled „Homology Modeling, Molecular Docking and Molecular Dynamics Simulation Studies of CALMH1 against Secondary Metabolites of Bauhinia variegata to Treat Alzheimer’s Disease”. Alzheimer's disease is the most common type of dementia, characterized by gradual brain cel loss and degeneration. Current drugs can only delay illness or may temporarily alleviate symptoms. Thus, effective therapeutic options are urgently need.

In recent times, diverse in silico software and algorithms to predict protein structure have been developed. The calcium homeostasis modulator 1 (CALHM1) is involved in the pathogenesis of Alzheimer's disease. Authors have presented created, by homology modelling and Rampage, useful models of the CALHM1 computational protein structure. In addition, research and analysis of the function of CALHM1 in Alzheimer`s disease were have been performed. Finally, quercetin was proposed as a suitable therapeutic inhibitor in treating this disease.

In my opinion, paper could be appealing for readers of Brain Sciences, and I recommend minor modifications. In particular:

- all figures (apart from 11) should be corrected due to poor quality (and labels are too small), the black background (Fig. might be changed to white

- redaction of paper (e.g. font, references, edition of texts, e-mail adresses of co-authors, subscripts/superscripts, etc. - section 2. Material and Methods should be bolded) should be carefully checked and corrected, according to Guides for Authors

-language could be corrected (e.g. line 25: „But in the absence… Through homology…; elimination of repetitions; line 85: These techniques, however, have limitations such as a high time). By the way, the latter statement is not adequate to reality. Nowadays, we have advanced automated apparatus and softwares, even for macromolecules.

Author Response

Answers to Reviewer 1

Reviewer 1

Comments

Explanation

all figures (apart from 11) should be corrected due to poor quality (and labels are too small), the black background (Fig. might be changed to white

Done

redaction of paper (e.g. font, references, edition of texts, e-mail adresses of co-authors, subscripts/superscripts, etc. - section 2. Material and Methods should be bolded) should be carefully checked and corrected, according to Guides for Authors

Done

line 25: „But in the absence… Through homology…; elimination of repetitions

Removed “But” from the sentence

line 85: These techniques, however, have limitations such as a high time

Added the term “requirement”

By the way, the latter statement is not adequate to reality. Nowadays, we have advanced automated apparatus and softwares, even for macromolecules.

Removed the statement

Reviewer 2 Report

Khare and co-workers reports the homology modeling, molecular Docking and dynamics simulations of CALMH1 protein to generate anti-Alzheimer’s agents. Although the manuscript details interesting results, the presentation and language of the article is extremely poor. The manuscript is full of linguistic, grammatical and technical errors. There are numerous mistakes in the WHOLE manuscript, and it is really hard to point out every single one. Some comments are given below.

Section 3.13. More discussion should be added to elaborate the results. The description about the molecular dynamic simulations is also very poor and no discussion on how and why the quercetin is showing interaction and how well it binds inside the active pocket. Please revise whole MD simulations section.

In the revised version I would also suggest to add the Figures of better resolution with readable amino acid residues (like Fig. 9a-d, 10, 12b are very poor).

General comments

Why compounds start with capital letter section 2.7 (line 153-157) and section 3.9 (line 281-283).

There are many typo and grammar mistakes in the manuscript e.g. line 359 there should be space between 10000ps. Please revise the whole manuscript carefully.

Same like previous 280 Pubchem should be PubChem…….

Authors have used many capital words within the sentences without any logic and scientific reasoning like line 39, 328, 339, 350, 360, 370, 385 and so on. It applies for the captions of Figures and Tables.

I wonder why authors have capitalize Q of Quercitin. I guess to emphasize on the results. It should be lowercase except in the beginning of the sentence.

Author Response

Answers to Reviewer 2

Reviewer 2

Comments

Explanation

In Figures 3 & 4, what are the different colors stand for?

The authors thank the reviewer for their comments. As per the suggestions, we have rectified in this revised manuscript.

In Figure 3

·         Dark blue is denoting Alpha helix.

·         Light blue is denoting Pi helix.

·         Dark pink is denoting Beta Bridge.

·         Red is denoting extended strands.

In Figure 4

·         Dark blue is denoting Alpha helix.

·         Green is denoting Pi helix.

·         Dark pink is denoting Beta Bridge.

·         Red is denoting extended strands.

What are the parameters used for energy minimization of the protein models? What is the requirements of this stage? One can run an MD simulation of the protein to equilibriate it in explicit water to check the fold stability.

The parameters that are used for energy minimization of protein molecule are by balancing the value of the machine by 40 ps when the temperature rises to 300 K. The simulations at 10 ns is performed at 1bar and at the temperature of 300 K.

The need for this stage was to find a set of coordinates representing the minimum energy conformation for the given structure.

SASA and radius of gyration has nothing to do with the accuracy of a simulation. These are the measures of stability of structure over the simulation time. There is fluctuations in the Rg plots for protein-ligand simulation (Fig. 14, 2500-10000 ps). Do not expect a change in SASA unless there is a significant structural change upon ligand binding, such as unfolding.

SASA and radius of gyration are the part of simulation although they show the stability of the structure over the time.

In SASA after binding up of Quercetin molecule with CALHM1, it is true that SASA graph should not expect any change but in the graph there is a bit declination which may be compared with the Rg plots of protein.

How many of the compounds here are blood-brain barrier permeable?

12

Please look in to the structures of the molecules used for docking. These are diverse class of compounds? Did all of them had the same binding site?

No, they don't all share the same binding site, however during docking, the best site binding was identified with Quercetin, since quercetin has the lowest binding energy.

Figurs 5 & 6 shoud be superimposed for comparison of the two structures. The same is true for other two models (Figures 7 & 8). That way one can see the regions of differences between different models.

The molecules which are selected from MODELLER and LOMETS server are different for the two templates (6VAM A and 6LMT A). The aim here was to screen molecule on the basis of DOPE and Z score respectively, not to see the regions in the protein molecule.

l. 32-33: Bauhinia varie-gata plant was used to check the interaction of alkaloids and flavonoids against CALHM1.

The alkaloids and flavonoids which are present in the bark of Bauhinia variegata plant were used for targeting the sites of CALHM1.

l.  32-33: Bauhinia varie-gata plant was used to check the interaction of alkaloids and flavonoids against CALHM1. How did the authors
use a "plant" in computation?

The alkaloids and flavonoids are retrieved from the Bauhinia variegata plant only.

l. 40: complex became stable at 2500 ps. Was the complex stable at 2.5 ns simulation frame, or throughout production simulation run?

No, the complex became stable at 2500ps while running.

l. 42: but maybe scrutinized for the best-predicted model. What did the authors mean by this statement? are their models not good enough?

Among different protein molecules, the best was screened for the further study.

l. 64: such presenilins, amyloid precursor protein...     Please add 'as' after "such"

The authors thank the reviewer for their comments. As per the suggestions, we have rectified in this revised manuscript.

l.67: Be consistent with the use of Ca2+o and Ca2+ o

The authors thank the reviewer for their comments. As per the suggestions, we have rectified in this revised manuscript.

l. 111: were also generated secondary protein [13].   What did the authors mean by "secondary protein" ?

The term “Structure” was missing from secondary protein which is added now.

l. 137: please replace "two template" with "two templates"

The authors thank the reviewer for their comments. As per the suggestions, we have rectified in this revised manuscript.

l. 145: What did the author mean by "projected structures and further structure"? Please define them.

It was mistakenly written as further structure which is replaced by further “study”.

Reviewer 3 Report

The manuscript by Khare et al. describe computational modeling of the metabolites of Bauhinia sp. as calcium channel inhibitors, as a
treatment to AD. The calcium hypothesis of AD is complicated in nature. It is not clear whether Ca ion dysregulation is one of the causes or effects of AD. The authors are not clear to the motivation of their work. What is the proposed mechanism of action of the CALMH1 inhibitors in AD treatment?

For docking runs, did the authors use the whole CALHM1 model(s) as search space or a specific binding site is targetted? In other words, was it a blind docking or a directed docking? What was the rational behind the use of two different docking codes?

The "Molecular dynamics simulation" section needs a total rewriting. The authors can follow any standard publication to find out the minimum required information to put here. In a related note, the authors used restraints for the distances using "all bonds" and as such they actually fixed the ligand position as obtained from the docking calculation. From these MD simulations one can not obtain stability of the protein-ligand complex. To do so, remove the restraints and run a longer simulation to see if the ligand stays in the binding site for an extended period of time.

In Figures 3 & 4, what are the different colors stand for?

What are the parameters used for energy minimization of the protein models? What is the requirements of this stage? One can run an MD simulation of the protein to equilibriate it in explicit water to check the fold stability.

SASA and radius of gyration has nothing to do with the accuracy of a simulation. These are the measures of stability of structure over the simulation time. There is fluctuations in the Rg plots for protein-ligand simulation (Fig. 14, 2500-10000 ps). Do not expect a change in SASA unless there is a significant structural change upon ligand binding, such as unfolding.

How many of the compounds here are blood-brain barrier permeable?

Please look in to the structures of the molecules used for docking. These are diverse class of compounds? Did all of them had the same binding site?

Figurs 5 & 6 shoud be superimposed for comparison of the two structures. The same is true for other two models (Figures 7 & 8). That way one can see the regions of differences between different models.

The manuscript has serious typo issues that make it difficult to follow. A few of the typos are as follow:
l. 32-33: Bauhinia varie-gata plant was used to check the interaction of alkaloids and flavonoids against CALHM1. How did the authors
use a "plant" in computation?
l. 40: complex became stable at 2500 ps. Was the complex stable at 2.5 ns simulation frame, or throughout production simulation run?
l. 42: but maybe scrutinized for the best-predicted model. What did the authors mean by this statement? are their models not good enough?
l. 64: such presenilins, amyloid precursor protein...     Please add 'as' after "such"
l. 67: Be consistent with the use of Ca2+o and Ca2+ o
l. 111: were also generated secondary protein [13].   What did the authors mean by "secondary protein" ?
l. 137: please replace "two template" with "two templates"
l. 145: What did the author mean by "projected structures and further structure"? Please define them.

Author Response

Answers to Reviewer 3

Reviewer 3

Comments

Explanation

Section 3.13. More discussion should be added to elaborate the results. The description about the molecular dynamic simulations is also very poor and no discussion on how and why the quercetin is showing interaction and how well it binds inside the active pocket. Please revise whole MD simulations section.

The authors thank the reviewer for their comments. As per the suggestions, we have rectified in this revised manuscript.

In the revised version I would also suggest to add the Figures of better resolution with readable amino acid residues (like Fig. 9a-d, 10, 12b are very poor).

The authors thank the reviewer for their comments. As per the suggestions, we have rectified in this revised manuscript.

Why compounds start with capital letter section 2.7 (line 153-157) and section 3.9 (line 281-283).

Converted all the compounds in lowercase.

There are many typo and grammar mistakes in the manuscript e.g. line 359 there should be space between 10000ps. Please revise the whole manuscript carefully.

The authors thank the reviewer for their comments. As per the suggestions, we have rectified in this revised manuscript.

Same like previous 280 Pubchem should be PubChem…….

The authors thank the reviewer for their comments. As per the suggestions, we have rectified in this revised manuscript.

Authors have used many capital words within the sentences without any logic and scientific reasoning like line 39, 328, 339, 350, 360, 370, 385 and so on. It applies for the captions of Figures and Tables.

Changed

I wonder why authors have capitalize Q of Quercitin. I guess to emphasize on the results. It should be lowercase except in the beginning of the sentence.

Changed

Round 2

Reviewer 3 Report

What does "SPBDV" stands for?

l. 278-279: what does the "best protein structure projected" mean?

In the Table 3 data, why the energy values are different, if it is for the same model? What are the reasons for such a high deviations?

l. 187: It is called the steepest descent algorithm.

Prodrg atomic charge assignments are known to be problematic. How did you verify the atomic charge assigned by prodrg to your ligands?

l.190-191: The use of bond restrictions negates the use of MD simulation to explore the stability of ligand binding. Please remove all restraints to see if you ligand stays in the binding site or it moves away. Such things are very common for docked complexes during MD simulation.  

Please use the subscript/superscript function of your document editor program for units. 

Author Response

Reviewer Comments: R2

What does "SPBDV" stands for?

Corrected to SPDBV (Swiss PDB Viewer)

What does the "best protein structure projected" mean?

Changed to “best protein structure which was predicted”

In the Table 3 data, why the energy values are different, if it is for the same model? What are the reasons for such a high deviations?

The reason for the high deviation is because of query cover and identity. In case of 6VAM A the query cover is near 99%and the value of identity is 68.36% while in case of 6LMT A the query cover is only 88% and the identity is 58.82%, due to which there is a deviation in values.

It is called the steepest descent algorithm. Prodrg atomic charge assignments are known to be problematic. How did you verify the atomic charge assigned by prodrg to your ligands?

Ligand atomic charges were computed via the restrained electrostatic potential and atomic charges were verified through Acpype gromacs.

The use of bond restrictions negates the use of MD simulation to explore the stability of ligand binding. Please remove all restraints to see if you ligand stays in the binding site or it moves away. Such things are very common for docked complexes during MD simulation

After removing the restraints, the ligand stays in the binding site only as shown in image.
